# A Qualitative Exploration of Fijian Perceptions of Diabetes: Identifying Opportunities for Prevention and Management

**DOI:** 10.3390/ijerph16071100

**Published:** 2019-03-27

**Authors:** Catherine Dearie, Shamieka Dubois, David Simmons, Freya MacMillan, Kate A. McBride

**Affiliations:** 1School of Public Health and Community Medicine, University of New South Wales Kensington Campus, Randwick, NSW 2052, Australia; 2School of Science and Health, Western Sydney University, Penrith, NSW 2750, Australia; 18381623@student.westernsydney.edu.au (S.D.); f.macmillan@westernsydney.edu.au (F.M.); 3School of Medicine, Western Sydney University, Penrith, NSW 2750, Australia; da.simmons@westernsydney.edu.au (D.S.); k.mcbride@westernsydney.edu.au (K.A.M.); 4Translational Health Research Institute (THRI), Western Sydney University, Penrith, NSW 2750, Australia

**Keywords:** diet, Fijian, lifestyle intervention, physical activity, qualitative, diabetes mellitus

## Abstract

Rates of diabetes are high in many communities of Pacific Island peoples, including people from Fiji. This qualitative study explores knowledge and attitudes towards diabetes among i-Taukei Fijians to facilitate the cultural tailoring of diabetes prevention and management programs for this community. Fijians aged 26 to 71 years (n = 15), residing in Australia, participated in semi-structured interviews; 53% (n = 8) were male. Interviews were audio-recorded, transcribed verbatim, then thematically analyzed. Diabetes is recognized as an important and increasing health problem requiring action in the i-Taukei Fijian community. Widespread support for culturally appropriate lifestyle interventions utilizing existing societal structures, like family networks and church groups, was apparent. These structures were also seen as a crucial motivator for health action. Intervention content suggestions included diabetes risk awareness and education, as well as skills development to improve lifestyle behaviors. Leveraging existing social structures and both faith and family experiences of diabetes within the Fijian community may help convert increased awareness and understanding into lifestyle change. Ongoing in-community support to prevent and manage diabetes was also regarded as important. We recommend building upon experience from prior community-based interventions in other high-risk populations, alongside our findings, to assist in developing tailored diabetes programs for Fijians.

## 1. Introduction

People originating from the South Pacific, including Fijians, are disproportionately represented in national diabetes statistics in Australia, where diabetes prevalence was estimated at 7.4% in Australians aged 25 years and over [1]. They are also more likely to be above a healthy weight [2,3,4] and are at higher risk of diabetes [1,5], with odds of diabetes being 6.3 and 7.2 times higher—after adjusting for age and socioeconomic status—for men and women born in the Pacific Islands compared to the Australian born population [2]. Diabetes-related hospitalization and mortality rates in Australia are 2.22 to 2.98 higher in South Pacific-born than Australian-born people [6]. 

Approximately 57,000 Fiji-born people live in Australia [7], with nearly 30,000 living in Sydney [8]. Predominant Fijian cultural groups are the indigenous i-Taukei Fijians and those of Indian ancestry (approximately 57% and 37% of the population in Fiji itself, respectively) [9]. These ethnic groups have distinct cultural, religious, and dietary practices [10,11]. This research focuses on the i-Taukei Fijian community (hereafter referred to as Fijian), approximately 13,500 of whom live in Sydney.

Culturally tailored diabetes prevention interventions have been shown to be effective in reducing risk factors among Pacific populations in New Zealand and in the USA [12,13,14,15]. There is a need for more culturally appropriate interventions to stem this ‘diabesity’ tide among specific Pacific groups, like Fijians. Perceptions of Fijians on diabetes and its risk factors could assist in culturally appropriate intervention development. This study aimed to describe Fijian cultural factors related to diet and physical activity in order to facilitate the development of appropriate and culturally tailored diabetes prevention interventions.

## 2. Materials and Methods 

One-on-one qualitative interviews (or couple focused where married individuals participated), were conducted to explore perceptions of Fijians living in Greater Sydney about the importance of diabetes in their community and how the disease might be prevented or better managed. Purposive recruitment was predominantly through a Fijian well connected within their community. A snowball method, through which participants were asked to invite others, was used to recruit further participants. Individuals were also invited at community events. Participants were required to be 18 years of age or older and to identify themselves as i-Taukei Fijian. Diabetes diagnosis was not a criterion for recruitment, but participants’ knowledge of diabetes could be expected to be influenced by their experience, either directly or indirectly, with the disease.

Guided by a schedule, the interviews explored perceptions about health, knowledge of diabetes, facilitators and barriers to preventing and managing diabetes, suggestions for interventions, perceptions of healthy weight, and readiness for change. The Interview Schedule was adapted from a guide used in a previous study in another population of Pacific Island origin in Sydney by the authors (F.M., K.M., and D.S.). It has not been previously published. (Appendix A). Photographic images [16] (validated for use in non-Pacific people for body image assessment) of females of various body sizes were used to prompt discussion around healthy weight perceptions (underweight, normal weight, and overweight).

Ethics approval (H12020) was received from Western Sydney University Human Research Ethics Committee (EC00314). Written informed consent was gained from all participants. All interviews were conducted in English by C.D. in quiet public places and were completed between January and April 2017. Interviews were digitally recorded then transcribed verbatim. Pseudonyms were used and identifying information removed from transcripts to ensure confidentiality. Occasional Fijian words were translated during transcription, guided by a Fijian community leader to ensure meaning was retained. Demographic details (age, gender, marital status, birth country, village or island in Fiji connected with, years in Australia, language spoken at home, education level, employment status, and diabetes status) were collected. 

Interviews were analyzed thematically, where data were systematically arranged into meaningful groups (themes and sub-themes) [17]. An initial coding framework was developed after four researchers’ (C.D., S.D., F.M., and K.M.) independently coded one transcript. After consensus was met on the coding framework, one researcher (C.D.) coded all transcripts. Independent consensus checking of 10% of all data was then conducted (F.M. and K.M.). The coding process used Quirkos 2.0 qualitative analysis software (Quirkos Limited, Edinburgh, Scotland,). A narrative around each theme is provided in the results with example excerpts referenced in square brackets—for example, [E1.2a]—and included in all the tables except in Table 1. In this example [E1.2a], “E” means excerpt, “1”is the first theme, “2” is the second sub-theme, and “a” is the first quote in that sub-theme.

## 3. Results

Fifteen participants were recruited, mean age 49 years (range 26–71 years) (Table 1). Two additional people had agreed to participate but both were shift workers. After several attempts to schedule a convenient time for interviews with these two potential participants, it was not possible without undue inconvenience for each of them and interviews were not pursued. In total, 15 people were interviewed, including three married couples who were interviewed as couples. A total of 13 hours of interview data was recorded and analyzed.

Two participants reported a diabetes diagnosis. Another nine had a first degree relative, and a further three had close relatives by marriage with diabetes. Only one participant did not identify a close relation with diabetes. 

Four overarching themes were identified: 1. Knowledge and awareness of diabetes, 2. culturally specific barriers to preventing or controlling diabetes, 3. structures that could be leveraged to prevent diabetes and its complications, and 4. recommended components of intervention.

### 3.1. Knowledge and Awareness of Diabetes 

Participants spoke about their knowledge of health issues including diabetes-related complications (Table 2). When asked about common health problems experienced by Fijians in Australia, several participants identified obesity (n = 6), diabetes (n = 9), and heart disease (n = 9) [E1.1a–1.1d]. Most participants stated they knew family members and friends diagnosed with diabetes [E1.2a and 1.2b]. Some participants demonstrated an understanding of the mechanisms and had general awareness of diabetes [E1.3a and 1.3b] but others were unable to define the condition [E1.3c] or believed there was a general lack of knowledge around diabetes in their community [E1.3d].

Participants were aware that diabetes is related to poor lifestyles [E1.4a] and appreciated the contribution lifestyle choices, including physical inactivity, had on health [E1.4b–1.4d]. More specifically, excess sugar and consumption of refined and processed foods were identified as primary causes of diabetes [E1.4e and 1.4f]. Others spoke about the level of health literacy in the community in relation to diet and its link to health. Participant perspectives suggested some members of the community had a reasonable understanding of the connection between health and diet [E1.4g–1.4i].

A third of participants had some existing knowledge about complications of diabetes, though, primarily, either they or a close family member had been diagnosed with diabetes. Knowledge level was not assessed in depth in this study. The most widely identified consequence of diabetes, and the one described as most impactful, was amputation [E1.5a–1.5c]. Other identified issues included social concerns such as needing to deal with medications, social isolation, burden on family, and an individual’s loss of ability to earn a living—no longer being able to drive a taxi due to foot amputation, for example. [E1.5d and 1.5e]. Overall, diabetes was recognized as a health concern for Fijians living in Australia and for family living in Fiji [E1.6a–1.6d].

### 3.2. Culturally Specific Barriers to Preventing or Controlling Diabetes

Possible barriers to a healthy lifestyle and consequently to diabetes prevention and management were discussed in the context of traditional diet and physical activity. Negative effects on these behaviors, such as dietary adaptation, transition to a more sedentary lifestyle, and acculturation were related to migration to Australia (Table 3). Many participants reported more disposable income since relocating to Australia, facilitating increased accessibility to larger quantities and convenient food options [E2.1a and 2.1b], including take-away foods [E2.2a–2.2c]. Difficulty in accessing healthier aspects of the traditional Fijian diet and lifestyle were identified by several participants as being a contributor to diabetes [E2.3a–2.3c]. Conversely, others described the traditional diet as being carbohydrate rich, with the addition of coconut cream common [E2.4a–2.4e]. Consumption of kava, a traditional drink with soporific properties made from the kava plant [18] and the social aspects of kava consumption were also observed as being a barrier to a healthy lifestyle [E2.5a and 2.5b]. 

Despite adaptations to the Australian way of life, other Fijian practices in food consumption have not been abandoned. The incapacity to practice portion control was a common comment [E2.6a–2.6d]. Family members’ influence could also inhibit healthy eating practices, with feasting at frequently held gatherings customary [E2.7a–2.7d] and failing to indulge seen as failing to partake in family life [E2.8a and 2.8b]. Furthermore, overindulgence was felt to be acceptable as medication use could compensate for this [E2.9].

A lack of awareness and understanding of diabetes was seen as a barrier for some in preventing and managing diabetes, including older people and those less health literate, as this could lead to avoidant behaviors such as not visiting a doctor [E2.10]. Avoidance could be exacerbated by traditional Fijian approaches to health through prayer and herbal remedies. This was seen as another potential barrier, as individuals may delay consultation with mainstream health professionals until they have tried this approach [E2.11a–2.11c] and then may not be compliant with taking medications.

Community perceptions of body image also appeared to act as a barrier to a healthy lifestyle. Being ‘curvier’ was perceived as attractive [E2.12a], and to lose weight or be ‘skinny’ was considered undesirable [E2.12b and 2.12c]. Babies that weren’t ‘chubby’ were seen to be undernourished [E2.12d]. When shown photographs of female body images, the most common pictures selected to represent a healthy body shape were those of people with an underweight BMI of 16.7 kg/m^2^ (n = 9) and 18.45 kg/m^2^ (n = 11) or normal weight BMI of 20.33 kg/m^2^ (n = 6). Conversely, when asked about what images would be considered an attractive body shape for Fijians, the answers were more varied, ranging from underweight to obese BMI values (18.45 kg/m^2^ up to 35.95 kg/m^2^).

### 3.3. Structures That Could Be Leveraged to Prevent Diabetes and Its Complications

As part of their lifestyle, Fijians are traditionally very communal [E3.1a and 3.1b]. Several participants said seeing their family members suffer was an important motivator for them to adopt a healthier lifestyle, and that leveraging this effect on family could be important in reinforcing the impact of diabetes [E3.2a–3.2c]. Including the whole family, particularly children, in any intervention approach was recommended as participants felt healthy eating practices, for example, should be instilled across the entire family [E3.3a–3.3c]. Table 4 records participant responses when asked for suggestions for encouraging adoption of a healthier lifestyle and how interventions could be implemented.

Others suggested another motivator for lifestyle change could be having a spiritual aspect to an intervention [E3.4a and 3.4b], particularly since church usually plays an integral part in much of the community’s everyday lives [E3.5a and 3.5b]. Church leaders are seen as very influential in the community, and possibly could be providing better examples of healthy eating [E3.6]. The social structures of family, church, and community were also seen as being appropriate to deliver interventions [E3.7a and 3.7b]. Community radio and social media were suggested avenues to raise awareness of diabetes [E3.8a and 3.8b], though the most frequent suggestion by which to do this was the Fiji Day community event, held annually in Sydney [E3.8b and 3.8c].

### 3.4. Recommended Components of Interventions

Education around nutrition was seen to be needed and specifically on portion size, food choice, and healthier food preparation methods [E4.1a–4.1d]. Several participants suggested that education around diabetes prevention should be confronting to get through to the community, i.e., by highlighting the likelihood that diabetes can have a large impact on their own health and the ‘realness’ of diabetes [E4.2a–4.2d]. Integrating physical activity into regular activities, like church attendance or work, was seen as a potentially useful strategy [E4.3a–4.3c]. Most participants highlighted the importance of involving the Fijian community to ensure cultural aspects are appropriate, otherwise there would be a risk of non-engagement in interventions [E4.4]. A number of participants suggested utilizing respected Fijian community members, such as athletes/sporting figures, nurses, or community leaders to help in intervention delivery [E4.5a–4.5c]. It was felt they would be more influential due to common shared experiences and could be “looked up to” as role models. Community members currently managing diabetes well were also suggested as potentially powerful mediums because they are real life advocates for diabetes management [E4.6a–4.6c]. Excerpts of suggestions for possible interventions are summarized in Table 5.

## 4. Discussion

The main findings of this study were the identification of social structures, such as family and church, as potential avenues for the delivery of an intervention. Furthermore, while a number of cultural barriers to diabetes prevention and management exist, there was much support for an intervention, (built upon social concern), an awareness of diabetes and its possible complications, and an understanding of the link between obesity and increased risk of developing diabetes. These data suggest that while awareness is not an issue, a lower level of knowledge exists about diabetes and its prevention, which is consistent with findings in other Pacific communities [19,20,21].

This community faces many of the same challenges as the wider Australian population in adhering to a healthy lifestyle for diabetes prevention and management, including social issues like lack of time due to long working hours, family responsibilities, and environmental issues such as accessibility to fast and processed foods [22,23,24,25]. Findings from our research suggest several factors specific to the Fijian community preventing their adoption of a healthy lifestyle. One of the challenges to addressing the likely higher rates of overweight and obesity and diabetes in this community is the cultural perception that being larger is an indicator of good health and social status [26]. While some participants in our study indicated this was not the case (when viewing body images), there were some contradictory responses to what is perceived as a healthy weight, suggesting a possible disconnect between recognition of a healthy body size and taking positive action to achieve this. For example, when seeing someone lose weight, our participants reported the typical reaction in the Fijian community was that the individual must be ill instead of recognizing they were improving their health; thus, losing weight was viewed negatively. A useful resource in helping communicate messages on body weight and health risk may be a culturally-tailored visual tool, such as a series of photographic images of Pacific body types illustrating increasing body fat levels which can then be linked to increasing health risk. Body image photos have been used previously among Cook Islanders who were asked to select the image that represented their own current size as well as the most healthy and attractive sizes for their own and opposite sex [27]. Female participants in that study were accurately able to identify their own current size with both sexes indicating that their preferred size was smaller than their own body size. Participants ranged in age in the current study (26–71 years) and their perceptions of body image may also be age dependent.

Migration to Australia is another important influence in this community which has led to increased sedentary behavior and limited access to healthier traditional diets. Acculturation has led to the combining of two worlds leading to perfect conditions to enable an unhealthy lifestyle. For Fijians, feasting is a significant part of many social activities and gatherings, where any attempts to practice appropriate portion control may be overwhelmed by other factors, such as bigger being seen as ‘better’ with respect to food portions, family and community influences on eating (where not eating is seen to be not participating), and cultural norms of body image. This is far from ideal when coupled with increased income, facilitating the ability to buy excessive quantities of food and easy access to cheap, fast, and processed foods. 

Given these cultural norms and practices, as well as ease in being able to adopt an unhealthy lifestyle, a whole community, culturally appropriate approach would appear to be the most practicable method for intervention in the context of the ongoing strong connections within the Fijian diaspora in Australia. Community based approaches can be achieved in several ways. First, based on Social Learning Theory principles [28], leveraging family and community experiences of diabetes may be a way to reinforce impact of the disease and as a motivator in adoption of a healthier lifestyle. This seems feasible given all but one of those interviewed had a close relative with diabetes. Second, as our data clearly conveyed that community—in particular family and church support—plays an integral role in everyday i-Taukei Fijian life; utilizing existing social structures like church groups could be a viable option for the delivery of a sustainable whole community intervention. This approach has been demonstrated as being effective in an intervention targeting Native Hawaiian populations in the USA and Pacific populations in New Zealand [13,15,29]. Furthermore, a recent study from the USA [30] illustrates how leveraging spiritual beliefs and practices in an African-American church based diabetes intervention may enhance health promotion and behavioral change. Third, any intervention should be culturally appropriate in consideration of established cultural norms and the spiritual nature of the community. Pacific churches play a significant role in the culture and authoritative systems of the communities that they serve, as well as providing a place for gathering and communication [31]. The adaption of existing cultural practices will likely be more realistic when aiming for adherence to healthier lifestyles, particularly as this approach has been shown to be effective in improving risk factors for progression to diabetes as well as diabetes management in minority populations elsewhere [32,33]. Fourth, working in partnership with the community to empower end-users to be involved in creation and implementation of health promotion-focused interventions in settings appropriate and attractive to the target group would ensure the uptake of programs [34] and long-term sustainability [35].

Cultural practices may also impact on access to health care, as traditional approaches to healthcare such as prayer and herbal medicines were mentioned as another possible barrier to diabetes prevention and management, which is similar to the findings of a study exploring barriers to diabetes care in New Zealand, including perspectives of individuals of Pacific origin, that also reported spiritual and alternative health belief barriers [21]. Members of the community engaging in these practices may be reluctant to connect with mainstream healthcare providers, as providers may be insensitive to traditional approaches if, for example, there is a lack of evidence to support safety and/or efficacy and the risk of interactions with herbal medicines [36]. From this perspective, too, the spiritual nature of the community as well as use of traditional herbal medicines should be respected, and interventions should be designed to incorporate these beliefs and customs. Integrative approaches incorporating treatment viewed through a cultural lens have been suggested to be successful previously in tailoring programs for mental health and substance abuse among Maori and Native Americans [37,38,39]. From the current data, it appears that there may also be others from the community who are not accessing health services, due to lower levels of health literacy, their learned experience growing up in Fiji where there is less accessibility to healthcare services due to limited availability and high expense, and being in denial about their health status. Consideration of access to health care and education of local primary healthcare providers should therefore be necessary elements of an intervention. Understanding the barriers to diabetes prevention and care can be used to create frameworks for healthcare providers to deliver support targeting barriers experienced by their patients [40]. 

Ensuring appropriateness of an intervention as well as leveraging of existing structures can also be achieved through involvement of community leaders like Ministers or lay leaders within churches in both development and implementation phases. By influencing the leaders in the community, there could be an established communication channel for disseminating information to the groups they guide. In Fiji, within the village setting, the Chief and the Minister would provide the day-to-day leadership to the community. Away from the village setting, members of the chiefly family still have an influential role in the community in other countries. Targeting these leaders first in an intervention so they can credibly support it will be essential. This is consistent with previous studies which explored community-engaged approaches and the influence of faith and tribal leaders in health-related issues [41]. Additionally, a number of participants suggested utilizing respected Fijian community members such as sporting figures, nurses, or community leaders to help in intervention delivery. These individuals may be influential in an intervention and potentially could act as role models and/or peer supporters for those taking part in an intervention. Peer support approaches have been shown to be effective in changing lifestyle behaviors, improving diabetes awareness, and reducing body weight in other culturally and linguistically diverse communities [42] as well as blood pressure and HbA1c improvements among individuals already diagnosed with Type 2 diabetes [43,44]. Additionally, a review of peer support interventions in New Zealand, including for Pacific groups, reported that peer support is useful [45]. This approach has been shown to be sustainable and cost effective. 

Intervention content should also be culturally appropriate. Several suggestions were provided by the individuals we interviewed. Surprisingly, despite lack of physical activity being explicitly linked to diabetes risk by our participants, there were fewer suggestions on how to include a physical activity element in interventions, with much greater emphasis given to dietary-related interventions. It seems feasible that group activities, such as exercise classes, walking groups, or cultural dance classes, could be conducted through existing church and community groups. These types of community based, culturally specific exercise activities have been shown to be effective in increasing physical activity [46,47]. Potentially these activities can also be led by peer supporters under the guidance of an exercise professional. Suggestions around nutrition education were based on the need for a whole community shift in attitudes towards food, which would be consistent with a whole community approach. Adaptation of traditionally consumed foods, education on the ‘right and wrong’ foods, healthy food quantities, and alternative food preparation methods were all specific ideas proposed by our interviewees. 

## 5. Strengths and Limitations

The semi-structured schedule used for the interviews was reviewed and tested with a member of the Fijian community prior to beginning the interviews to ensure cultural sensitivity and to ensure questions would be interpreted correctly. Our sample may not be representative of all Fijians in Sydney. The age range was heavily weighted to the 45–50-year-old age bracket, which may represent people who are more interested or more aware of diabetes, as it is more common as people age. However, the ages of participants did range from 26 to 71 years, ensuring representative input across the adult age range including the 65 years and over group, which accounts for only 9% of the Fiji born population in Australia [7]. Furthermore, the education level of our participants was high, and all spoke English well. This may have biased our data as those with lower education levels and who may experience language barriers when accessing healthcare could have lower levels of diabetes knowledge. However, less than 3% of Fiji-born people in Australia report speaking English not well or not at all [7]. All but one participant had a close relation with diabetes. Thus, sampling bias may have been evident as participants may have been motivated to participate due to their connection to someone already with the condition. Qualitative research does not aim to generalize; rather, it aims to explore the possible topics that are important to the target group. However, we recognize that perceptions may differ in individuals with different characteristics to the current sample. 

The interviewer was Australian but has had extensive interaction with Fijian communities over the last 7 years. Though she has developed a level of appreciation of the Fijian culture, she may have missed some information that a Fijian would pick up on. 

## 6. Conclusions

The data collected in this study in addition to the existing literature indicates that lifestyle interventions promoting a healthy diet and increasing physical activity for Fijians should be cognizant of their cultural values and practices. Interventions should incorporate Fijian cultural content. Leverage of family and church support would enable culturally acceptable and sustainable approaches. The supportive environment of a church community may be particularly suited to preventing and controlling diabetes in Fijians who are living away from the village structures in Fiji. Our findings have allowed us to develop a set of recommendations for future intervention development in this target group (Table 6).

## Figures and Tables

**Table 1 ijerph-16-01100-t001:** Demographics of Adult i-Taukei Fijians interviewed (n = 15).

i-Taukei Characteristics	Male, n = 8		Female, n = 7	
	Range	Mean	Range	Mean
Age (years)	36–71	50	26–55	46
Years in Australia *	7–37	24	7–24	16
**Marital status**				
Married	6		6	
Single or widowed	2		1	
**Birth country**				
Fiji	7		7	
Australia	1		-	
**Diabetes diagnosis**				
Yes	2		-	
No	6		7	
**Highest level of education**				
University	4		4	
Certificate/Diploma	4		2	
High school	-		1	
**Employment status**				
Full time	7		6	
Part time	-		1	
Retired	1		-	

* One female interviewed normally resides in Fiji, so years in Australia would be 0. This participant is not included in summarizing “years in Australia” but is reported in other demographic date.

**Table 2 ijerph-16-01100-t002:** Excerpts for knowledge and awareness of diabetes and health.

Sub Theme	Example Excerpts
**Knowledge of common health problems**	E1.1a: The most common one that they would have is, um is obesity. Obesity. Along with obesity, one that is common is the kidney, kidney dialysis…one that follows with heart problems is um stroke. (50 years old, male)
	E1.1b: Lifestyle, especially heart diseases…yeah, NCD’s, non-communicable diseases. So they call it, lifestyle diseases. One of them is diabetes. (45 years old, male)
	E1.1c: One, number one is diabetes. A lot of our families and friends now have that…I would say heart, number two. Yeah, cholesterol too. (49 years old, female)
	E1.1d: The main common things that I know for Fijians, either they got diabetes, high blood pressure or heart [disease]. (47 years old, male)
**Diabetes in the family**	E1.2a: A lot of our families and friends now have that [diabetes]. Ahh people I’m close to yeah. With um, that have ahh can say maybe five I can name with my... the top of my head who are now taking insulin injections. (49 years old, female)
	E1.2b: So, I think that’s the thing, yeah, I’ve got lots of family that have diabetes. (36 years old, male)
**Understanding of diabetes**	E1.3a: I know diabetes is when your body does not process sugar in the way that it should. So, it produces, so it doesn’t process insulin, um in the way that it should. (48 years old, female)
	E1.3b: We have a health and wellbeing [missing word] at work and they come and talk to us about um eating healthy and what type 2 diabetes is and how it can be managed yeah…looking for type 2 diabetes, is your diet and your exercise. And that’s the way I can treat it, um to manage it. (47 years old, male).
	E1.3c: No, not technically, I just sort of know it as like a, like the way it’s used and thrown around and I know the really bad stereotypes. (26 years old, female)
	E1.3d: But I think knowledge is very limited in regard to what it actually... how it manifests and how it actually is ahh represented in the way in which people ahh live healthy lifestyles either type 1 or type 2. (36 years old, male)
**Causes of diabetes**	E1.4a: Type 2 is more connected to your health. Well both are connected to your health, but people may have type 2 as a result of their diet and nutrition…It’s something that is preventable; that it does impact your wellbeing. It is related to weight and obesity. (36 years old, male)
	E1.4b: Lifestyle, I think is really important, food is for me, food is your 80% there. And then the other 20% is exercise…the culture aspect is around habits that have been passed down. And that is: unhealthy diet, overconsumption, choices of food. (46 years old, male)
	E1.4c: Poor diet, not a lot of exercise…Everything contributes to healthy living, like exercise, it’s not only diet, because your, your activity that you do; there’s so much more than just diet contributes to having diabetes. (50 years old, female)
	E1.4d: Yeah, it’s the ahh like I said the lifestyle um no exercise, lack of sleep, eating um different times of the day. (49 years old, female)
	E1.4e: So, I would say eating lots of refined things, um an imbalance in the amount of sugar you’re taking in and burning off and the types of sugar probably. (26 years old, female)
	E1.4f: That’s what causes it, like too much carbohydrates, that’s sugar. (71 years old, male)
	E1.4g: But that [diet] contributes to our health, that we lack to understand that it does contribute to our health. (45 years old, male)
	E1.4h: Um, I think there would be very few people who would look after their diet, very few I would think. So, yeah, if there is awareness, would that change it? (48 years old, female)
	E1.4i: I mean I don’t touch McDonald’s or KFC, I ahh, I just can see it’s, it’s just a food that’s gonna kill us. (49 years old, female)
**Consequences of diabetes**	E1.5a: Some people can get amputated, amputation that’s all I know. (47 years old, male)
	E1.5b: The consequences of diabetes, is the family. Like my aunt, my uncle was amputated. He was in a wheelchair. (45 years old, male)
	E1.5c: Yeah amputation, people feel isolated, psychologically because they know they’ve been you know, they got diabetes. (45 years old, male)
	E1.5d: You have to then take medication for having to look after your condition…the impact on your quality of life. (46 years old, male)
	E1.5e: I’ve got two uncles there [Fiji], they’ve got diabetes. Yeah, they’ve amputated their foot. And my brother’s got diabetes. (50 years old, male) (his uncle, he was a taxi driver (49 years old, female)) …cos they amputated his leg…he can’t drive. (50 years old, male)
**Diabetes is a major health concern for Fijians**	E1.6a: Diabetes is quite common. Quite common like in Fijian community [laughter]. I know there’s a lot of yeah diabetes. (47 years old, male)
	E1.6b: Cos diabetes right now, it’s so common in Fiji that I am sure, like most Fijians here would have relatives back home who have diabetes. (49 years old, female)
	E1.6c: It is [an] epidemic [diabetes], it is a big concern. (46 years old, male)
	E1.6d: [health problems in Fiji], they’re changing. And the lifestyle, one of the areas, one of the diseases is now growing is heart disease. And heart disease comes with diabetes, so these are two good friends. And they’re normally associated with lifestyle, the heart cases and diabetes. (45 years old, male)

**Table 3 ijerph-16-01100-t003:** Excerpts related to culturally specific barriers to preventing or controlling diabetes.

Sub Theme	Example Excerpts
**Higher financial income increases ability to buy food**	E2.1a: It’s easy to go and buy a good cut of steak, that you’ll have like a big, you have a steak each, for everyone. So, there’s more meat…and only tiny salads. So, because there’s money, you can buy, yeah. (48 years old, female)
	E2.1b: …and so instead of cooking like a pot of fish, which is probably good, you know, we’ll go get KFC, and we’ll go get pizza because it’s available, right. Because now we have the money. Like in Fiji you don’t have that kind of money. (48 years old, female)
**Availability of fast food**	E2.2a: I think that we have more ready access to other foods, um and fast foods, which again I think is an issue in itself. (36 years old, male)
	E2.2c: But for Australia, you know, especially we have the freedom of choice and the busy life, we tend to convenience, so the food is available in supermarket, why not. The food is readily available, and all those food outlets, like fast food, why not. So, the availability changes people’s lifestyle. Or convenience, because everybody’s rushing. (45 years old, male)
	E2.2c: And so instead of cooking like a pot of fish, which is probably good, you know, we’ll go get KFC and we’ll go get pizza because it’s available, right. (48 years old, female)
**Traditional diet being healthier**	E2.3a: So, when I say culture, I don’t mean in terms of traditional Fijian culture because that actually is the savior here. So, when I go to Fiji, I eat all the traditional foods, which is unprocessed, low sugar, living off the land, you know. (46 years old, male)
	E2.3b: There is, still the, we still are having our sort of Fijian diet. Like mainly root crops and ah, the greens, like taro leaves and ah, bele and all these things. (71 years old, male)
	E2.3c: [In Fiji] Most people eat fresh foods, either from the farm or from the sea. And um they cook their own food, so they know exactly what’s going into the pot. In here [Australia] they really don’t know, they only get a container or a plate so they really don’t know what are the ingredients. (45 years old, male)
**Unhealthy traditional diet**	E2.4a: Lots of starches. Lots of carbs, so carb loaded. Rice, dalo. Dalo is our taro. Cassava, um, bread, it’s heavily, I mean its readily available, right. (48 years old, female)
	E2.4b: We like layer things in um coconut cream and coconut sauce um and then we have lots of starches like taro and cassava, we love them. (26 years old, female)
	E2.4c: Starchy food and, and, and people like to have fish ahh food cooked in coconut milk. (47 years old, male)
	E2.4d: Um we eat a lot of taro and cassava um roti, curries from an array of different meats, rice, coconut everything and then some. (36 years old, male)
	E2.4e: Very carb loaded. And very protein full as well, we eat a lot of meat. (48 years old, female)
**Kava is consumed frequently**	E2.5a: I think mainly because people drink too much kava. (71years old, male)
	E2.5b: That’s one of the effects—taking time off doing other things: sleeping, eating… because if you drink kava, you’ll spend a lot of time, so you’ll miss taking your tablets and you’ll go and eat late at night and then you’ll sleep. (55 years old, female)
**Fijian food consumption practices**	E2.6a: Let me tell you, we don’t know what moderation is really, we go “full on”. We love our food. We love our food. (48 years old, female)
	E2.6b: I think we need to be informed, how much is enough for you for the day, you know. Like, basically you don’t need all this, this is sufficient to hold you until your next meal, for your body to function. (48 years old, female)
	E2.6c: Cook them in small amounts if amounts is a problem for you, which it is for Fijians, portions are... Fijians don’t practice portion control. (26 years old, female)
	E2.6d: The main thing in the Island thing is, everything’s the food, it shows how wealthy to you are… When you have a function the first thing you’ll think is you know, don’t run out of food, so you gotta make sure everyone eats. [laughter] (47 years old, male)
**Family and community influences**	E2.7a: So, and but no one stops to think I shouldn’t eat that, you’re like your all together there’s like a family mentality built around food so to not eat food is like to not participate in the life of the family at that moment. (26 years old, female)
	E2.7b: You know, eat. Why? You don’t like our food? You know, you’re offending me because you’re not eating my food. What’s wrong with you? (48 years old, female)
	E2.7c: It’s all these communal functions that you go to, gatherings, we have a lot of feasting, you know, feasting, eating at the wrong time. That’s, that’s ahh, you know Fijian—or Pacific Island functions are big you know. When we have a function it’s a big feed. (47 years old, male)
	E2.7d: When you have a function the first thing you’ll think is you know, don’t run out of food, so you gotta make sure everyone eats. (47 years old, male)
**Family pressure to eat**	E2.8a: …there’s like a family mentality built around food, so to not eat is like to not participate in the life of the family at that moment. (26 years old, female)
	E2.8b: like today’s gathering, if there’s food I’m not supposed to eat, but because I love to indulge with my family, I’ll indulge in it. I know it’s not good for me. But I’ll still indulge in that. … But you know, I don’t want to miss out. (45 years old, male)
**Medication**	E2.9: …you know the mentality of “the medication will do the job, I can eat how I eat. Medication is there to do its job. (55 years old, female)
**Lack of awareness and understanding of diabetes**	E2.10: If they’re not aware, it’s got no control…the elders is frightened to go to the doctors for medical check-up, that is their weakness… some people are scared to know what kind of sickness they have and or early signs. (47 years old, male)
**Spirituality and Traditional approaches to healthcare**	E2.11a: This is the challenge, we have religion, we have our Fijian herbal medicine and then we have the doctor……we’ve been brought up with that belief. Go to church. This is our cultural thing, we believe in this, and these two are intertwined. (prayer and herbal) (45 years old, male)
	E2.11b: And the thing with us, like tablets and other medications is not our life, like we’ll resort to herbal and that’s it. But it has to become really serious, you get pains and you are serious, then you’ll follow the doctor’s orders to take it because you’re aware that you don’t want to suffer the pain and everything else. That’s lack of education for us. (50 years old, female)
	E2.11c: Also people, when they got sickness, like they don’t want to say, it’s very private …if you have a problem, it’s in our culture that you want to solve it yourself. You either go and ask somebody quietly, or go herbal medicine, or you know, you wanna do on your own, rather than going straight to the solution. (45 years old, male).
**Cultural perceptions of weight**	E2.12a: We don’t have body shaming as an issue. Just because you’re curvier, even if you are this (indicating larger body images), you can… still seen as attractive. (46 years old, male).
	E2.12b: In a typical Fijian mind, if you’re losing weight then somethings wrong with you. It’s not like you’re trying to be health conscious, it’s you’re, like you’re just sick…I’m thinking of a traditional mind, right, the rounder, for women, that’s more attractive. (48 years old, female)
	E2.12c: Bigger, yes, yes, prefer bigger. Because to us, if you’re skinny like this (indicating the smaller body images) they think, oh, what is he feeding his wife? Because not enough food to—if you pile on food, they bigger, you know. He’s looking after his wife. (64 years old, male)
	E2.12d: If your child is undernourished, you’re not a good parent. You need to have a chubby baby to be a real pacific baby. (55 years old, female)

**Table 4 ijerph-16-01100-t004:** Excerpts related to facilitators for preventing diabetes and its complications.

Sub Theme	Example Excerpts
**Maintenance of family, faith and community connections**	E3.1a: We are people who gather as well; we are very communal people. (48 years old, female)
	E3.1b: I would do workshops and introduce to them new, um, using our own food crops but you know modern recipes for girls…I think we should be educated. And ah, in the community, to help one another, and do it as a group because we are communal people. (50 years old, female)
**Leveraging connections to family to highlight impact of the disease**	E3.2a: And like, if they can relate to someone close to them who has diabetes, you know, that is an encouragement for them to straighten out their lives. Well it’s, well that’s the case with me. You know, that my mum. So like when I slacken off my exercise, when I think about that, that makes me go out and… (49 years old, female)
	E3.2b: In dispersing that information I would take a really heavily family based stance, where I would outline it means early death if you’re a father, for your child it means this cause you’ll pass down your eating habits, um what it means for your wife, cause that’s what Fijians listen to... the effect that it’ll have on their families, even, not themselves. (26 years old, female)
	E3.2c: Run specific campaigns on educating family members, and their role and responsibility in helping their family member with diabetes would be helpful. As you know Pacific including indigenous Fijians are very communally orientated so having that context, I think would be helpful. (36 years old, male)
**Whole of family approach**	E3.3a: You have to actually encourage their families, you know, to support them, you know. Maybe eat the right food that they supposed to eat…Like how this thing, how it actually starts. Cos I think, like diabetes, it doesn’t start like just overnight. It takes a while. Yeah, so I think, educating even like our young kids, very very important (49 years old, female)
	E3.3b: Um, you know education in schools. Programs in schools. I don’t know, check lunch boxes and things like that. (48 years old, female)
	E3.3c: Because you know, it’s going to be really hard for them eating those, sometimes tasteless food which is supposed to be good for them, while the rest of the family is, you know, enjoying those taro, pork and all those fatty things. (49 years old, female)
	E3.3d: You are able to speak to them and encourage them, that you preparing what… its and teaching your children good health habits, good foods. Preparing good lunches. (55 years old, female)
**Spiritual connections**	E3.4a: I would probably throw something spiritual in there, you know how the bible says to take care of your whole self, not just turn up to church and do these things (26 years old, female)
	E3.4b: Coming from a biblical perspective, from a Christian world view. Yeah… Yeah, that would be a good way. Oh, start off with Biblical scripture…. Away we go. (50 years old, male)
**Church is integral in everyday life**	E3.5a: Especially more so if they’re going to a church where the Fijian community is, so if there’s a strong Fijian cultural community there’s heaps of loyalty to that, to that priest and that church and I guess that priest would exercise some sort of influence. (26 years old, female)
	E3.5b: Churches yeah cause that’s a good number [of people]. Everyone goes to church. [laughter] (49 years old, female)
**Church leading by example**	E3.6: First of all it’s the church leaders them self they should be worried about their diet, that, they are the, if you look at the some of these preachers and some of these priests and all, their lifestyle is not good so they’re the first person who should be, you know, as an example. (47 years old, male)
**Social structures for intervention delivery**	E3.7a: One is through church, one is if there’s ahh through family functions, relatives, yeah family functions, um or friends, just normal Fijian friends meet together…And what we just try, it’s more talking about, instead of talking about a useless thing we’ll talk about ahh our ah health and wellbeing. Then maybe we can pass on just by all while drinking kava, instead of talking nonsense people talking about this. (47 years old, male)
	E3.7b: You need to go to the church elders. Then the church elders will take it down to their little cell group meetings. Then from the cell group meetings, they’ll take it down to their families, to their children…or you can take it to the youth group meetings. (50 years old, male)
**Raising awareness of diabetes**	E3.8a: Also, the use of you know, social media…. Either a newspaper, television or, there’s supposed to be a lot of promotion on ah diabetes. What are the ways to detect it, what are the solutions there. Who you can visit. The steps to avoid it. (45 years old, male)
	E3.8b: One of the biggest gatherings is Fiji day, it gathers about close to 10,000 people. That’s an annual thing…there are a couple of Facebook pages, Sydney Facebook pages. There’s a few radio stations…because the older people will be listening to Fijian radio, right. The younger people, social media. (48 years old, female)
	E3.8c: One of the most well attended community event in Sydney is Fiji day, where it’s now … grown to be more than just Fijians which is great… it’s an opportunity to definitely um run those sort of stalls not just to give out pamphlets but also to encourage people to have a conversation. (36 years old, male)

**Table 5 ijerph-16-01100-t005:** Excerpts related to recommendations for components of an intervention.

Sub Theme	Example Excerpts
**Nutrition advice**	E4.1a: Um for the younger generations definitely looking at the nutrition stuff that’s in the context of [excuse me] healthy living. (36 years old, male)
	E4.1b: Similar to that you can just have foods that are good for you and stay away from, I went to the hospital also last month to visit my cousin who has diabetes. They had a stall there and all the foods, all the drinks that to be avoided, they had it all there on the table and the good foods. (49 years old, female)
	E4.1c: We need to be informed, how much [food] is enough for you for the day, you know. Like basically, you don’t need all this. (48 years old, female)
	E4.1d: Like we’ve given you suggestions of what you can eat and things that you guys actually like, maybe just cook them a different way or cook them in small amounts if amounts is a problem for you. (26 years old, female)
**Scare tactics to raise awareness**	E4.2a: You have to—sometimes, my people, the Fijians, they are a tough nut to crack. So some of times you have to scare them with the words. (55 years old, female)
	E4.2b: I think it’s just tell them straight, the consequences you’re gonna have. (64 years old, male)
	E4.2c: Maybe some drastic thing, so if you want to eat taro, this is what happens. Yeah something like that, that can attract attention because the passive approach won’t happen, won’t change minds. We’re too comfortable. (48 years old, female)
	E4.2d: We know all that sort of stuff, so I think it’s more around education programs around the *realness* of diabetes. (36 years old, male)
**Increasing physical activity**	E4.3a: …well, you know, be more active around the house. Cleaning and you know [laughing]. (50 years old, male)
	E4.3b: Like, back to the church, incorporate that, you know, into the church. Where maybe within the ladies group, to encourage them to…(49 years old, female)… Like in the morning before they go, you know, after their prayer meeting, go for a walk: One hour walk, half an hour, 45 minutes. Go to the park… Um. Take the kids with them, go walking with the kids. (50 years old, male)
	E4.3c: Oh, probably just a simple, like we have at work, a pedometer challenge… but by having something simple as walking, with a pedometer, you not only recording it, you using your smart phone... So by doing that, everyone will be competing. Oh look, I beat you, I beat you. (45 years old, male)
**Common understandings**	E4.4: Some people will look at things, they will go, oh yeah, but that’s a white person. You know, they don’t know my lifestyle. So then they just, that’s how you explain it away. But if you can then bring somebody to them, and have a conversation that they understand, about things that matter, then they will see if. Otherwise it’s foreign, right. (46 years old, male)
**Community involvement in intervention**	E4.5a: Fijians in Australia, just by my own experience probably find influence in other Fijians who seem to make it in the community, so sports star; rugby league, rugby union, um boys that make it to that level are really heralded as successes and things to look up to. (26 years old, female)
	E4.5b: Invite a nurse from Westmead, the Fijian nurse from Westmead, just to come and talk about health issues …get someone, a professional from there, a dietitian who’d understand our food, and you know, can talk about it. (48 years old, female)
	E4.5c: But there are, um, there are some community leaders as well, who are nonreligious. But it’s probably because either their political affiliation or a chiefly status, that will make them be seen as a leader. (48 years old, female)
**Community advocates and peer support**	E4.6a: You know, can use people that have gone through taking medication and talking from experience. Because some people will say, oh I don’t wanna take this medication because blah blah. You know, convincing other people not to take, oh you take other things. But then this person would say, you know, I’ve lived with diabetic and it has controlled my sugar like for 50 years, in fact, if I can do it, you can do it. Use those resource persons and plus those victims. (50 years old, female)
	E4.6b: I think the best thing is its coming from each individual, to share. Because most of our families here are suffering the—they share what advice they can give…it’s [more] powerful than reading books. Than who is not a sufferer. (64 years old, male)
	E4.6c: People have to see things to get motivated. If they just hear it they say “oh, what is this?” So, when they see things that’s goanna probably cause them some issues, people tend to act. So, for us, the act is the problem. (45 years old, male)

**Table 6 ijerph-16-01100-t006:** Intervention suggestions to prevent, and improve management of, diabetes.

Suggestion	Explanation
Awareness raising related to diabetes and obesity	Utilize Fijian radio, Fijian Facebook pages and a stall at Fiji Day in Sydney. Develop messages in consultation with influential leaders from the Fijian community. Images of healthy body weight i-Taukei, or other indigenous Pacific Islanders, could help people to self-assess their own risk more realistically and objectively, rather than comparing within their community or to those in Fiji.
Education	Provide general health awareness education to increase health literacy. Provide more in-depth understanding of diabetes: How it develops, can be prevented and managed, and potential complications. Endorsement needed by community leaders, in particular church leaders. Include information related to improved diet and increased physical activity. Ask those currently living with diabetes to share their experiences: People whose diabetes is well controlled as well as people who have experienced complications.
Religious and family context	Incorporate the Fijian approach of prayer, herbal and medicine into education sessions. E.g., church leaders could introduce sessions with relevant bible passages to provide motivation for behavior change. Highlight potential impact on families of a diagnosis of diabetes.
Skill development suggestions	Cooking workshops demonstrating healthier food options and cooking techniques, tailored to the Fijian palate. Use ubiquitous communal lunches after Sunday church services as an opportunity to present healthier food and introduce it to a wider group to support community wide change. Provide support for growing vegetables at home or in a community garden which could be effective given the communal nature of Fijian people. For people living in units, planter boxes that can be set up on balcony spaces could be provided with seeds to grow vegetables, and instruction on how to set this up and keep it going.
Cultural activities with a twist	Incorporate regular sharing of diabetes or health-related information during informal chats ‘around the kava bowl’, rather than just talking “nonsense”. Identify community champions to initiate the process, to raise awareness about diabetes and its complications and allow sharing of what they have found to be effective ways to manage the challenges of their condition. Facilitate a trained coach from the community to support community champions.

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
