# Peer review of "A Qualitative Exploration of Fijian Perceptions of Diabetes: Identifying Opportunities for Prevention and Management"

_ijerph, 2019, doi:10.3390/ijerph16071100_

Round 1
Reviewer 1 Report
This manuscript describes a qualitative analysis of semi-structured interviews toward cultural tailoring of interventions among the Figian community, particularly the i-Taukei Figians, in Australia. This is a worthy contribution to the literature, as the authors’ experience is valuable to others who are doing cultural adaptations, particularly for Figian and other Pacific Islander populations. Attention to the Figian community has been underrepresented in this literature. The paper is well written and the study was thoughtfully designed and implemented. My comments relate to only a few areas were the authors might be more clear.
While the paper’s title and text indicate the purpose is to explore intervention opportunities for prevention and management of diabetes, the abstract—in 2ndsentence—indicates diabetes prevention only as the purpose. This is somewhat confusing.
The authors describe their snowball approach to recruitment, which yielded 15 individuals. However, there was no mention of eligibility criteria other than being of i-Taukei background. Participants did not all have diabetes or have a family member with diabetes; one had neither. Perhaps this person was justified as all in the community were at risk, but this was unclear. Was the sample size of 15 the target number of participants? This sample is somewhat small, even for qualitative research, but it could be justified if the sample represents the community that was targeted. It is unclear what population features were sought in recruitment. Did all of the individuals approached agree to participate? Or, were there any refusals that should be reported?
I found the participant’s knowledge about diabetes and diet to remarkable and their vocabulary was quite sophisticated. But this observation was appropriately acknowledged in the limitations section as a possible source of selection bias.
The discussion (bottom of page 12) noted that participants made no suggestions on physical activity, however I read one suggestion on page 9, section 3.4, line 173, “integrating physical activity into regular activities, like church attendance or work…” Yet indeed, there was minimal attention to this topic.
I thought the discussion did a very good job of reviewing the literature on cultural adaptation of diabetes interventions in order to reinforce suggestions from participants in this study.
Author Response
Reviewer: This manuscript describes a qualitative analysis of semi-structured interviews toward cultural tailoring of interventions among the Figian community, particularly the i-Taukei Figians, in Australia. This is a worthy contribution to the literature, as the authors’ experience is valuable to others who are doing cultural adaptations, particularly for Figian and other Pacific Islander populations. Attention to the Figian community has been underrepresented in this literature. The paper is well written and the study was thoughtfully designed and implemented. My comments relate to only a few areas were the authors might be more clear.
Response: Thank you for your supportive comments.
Reviewer: While the paper’s title and text indicate the purpose is to explore intervention opportunities for prevention and management of diabetes, the abstract—in 2ndsentence—indicates diabetes prevention only as the purpose. This is somewhat confusing.
Response: Thank you – this has been corrected in the abstract. The abstract now reads:
This qualitative study explores knowledge and attitudes towards diabetes among i-Taukei Fijians to facilitate cultural tailoring of diabetes prevention and management programs for this community
Reviewer: The authors describe their snowball approach to recruitment, which yielded 15 individuals. However, there was no mention of eligibility criteria other than being of i-Taukei background. Participants did not all have diabetes or have a family member with diabetes; one had neither. Perhaps this person was justified as all in the community were at risk, but this was unclear. Was the sample size of 15 the target number of participants? This sample is somewhat small, even for qualitative research, but it could be justified if the sample represents the community that was targeted. It is unclear what population features were sought in recruitment. Did all of the individuals approached agree to participate? Or, were there any refusals that should be reported?
Response: As this was a qualitative study, the sample was purposively sampled. The criteria for recruitment were:
Adults aged 18 years or above
I-Taukei Fijian & residing in Sydney
With or without diabetes
The following new text has been added at line 60:
Participants were required to be 18 years of age or older who identified themselves as i-Taukei Fijian. Diabetes diagnosis was not a criterion for recruitment, but participants knowledge of diabetes could be expected to be influenced by their experience, either directly or indirectly, with the disease.
Diabetes prevalence in adults in Australia is reported to be around 6% (in the total adult population) while in Fiji it is around 16.6%. Specific data on the prevalence of diabetes in Fijians living in Australia is not known. However, from our work with other Pacific communities, the prevalence of diabetes is likely to be considerably higher than 6% in Fijians living in Australia. The participants referred extensively to family members who remain in Fiji, where diabetes prevalence is much higher than in the general population in Australia.
The initial target sample size was around 20. However, with preliminary analyses of the first 15 interviews completed, saturation was attained in terms of the feedback being provided. These 15 interviews yielded around 13 hours of recorded data.
There were no refusals to participate. There were however two additional people who had agreed to participate but both were shift workers and subsequently were unable to schedule an interview time After several attempts to schedule a convenient time for interviews with these 2 potential participants, it was not possible without undue inconvenience for each of them and interviews were not pursued. Text describing participants has been expanded as follows:
Fifteen participants were recruited, mean age 49 years (range 26–71 years) (Table 1). Two additional people had agreed to participate but both were shift workers. After several attempts to schedule a convenient time for interviews with these 2 potential participants, it was not possible without undue inconvenience for each of them and interviews were not pursued. In total 15 people were interviewed, including 3 married couples who were interviewed as couples. A total of 13 hours of interview data was recorded and analyzed.
Reviewer: I found the participant’s knowledge about diabetes and diet to remarkable and their vocabulary was quite sophisticated. But this observation was appropriately acknowledged in the limitations section as a possible source of selection bias.
Response: All but one of the participants who volunteered had a close family member diagnosed with diabetes. Having a family member with diabetes may have motivated some people to participate in the study and may have biased the sample. However, since this was a qualitative study it was not intended to be representative of the entire population of i-Taukei Fijians in Australia, but rather it focused on gaining richer responses specifically targeted at preventing and better managing diabetes in this community. As we had mentioned the limitations in regard to the selection bias, which the reviewer pointed out, we have not made additional changes to the manuscript.
Reviewer: The discussion (bottom of page 12) noted that participants made no suggestions on physical activity, however I read one suggestion on page 9, section 3.4, line 173, “integrating physical activity into regular activities, like church attendance or work…” Yet indeed, there was minimal attention to this topic.
Response: Thank you – the text now at the top of page 13 (line 302) has been corrected to more accurately reflect that participants offered less feedback related to physical activity interventions and tended to put much greater emphasis on dietary aspects to interventions. However, as the reviewer has noted, there were some limited physical activity related suggestions. Text has been amended as follows:
Surprisingly, despite lack of physical activity being explicitly linked to diabetes risk by our participants, there were fewer suggestions on how to include a physical activity element in interventions, with much greater emphasis given to dietary related interventions.
Reviewer: I thought the discussion did a very good job of reviewing the literature on cultural adaptation of diabetes interventions in order to reinforce suggestions from participants in this study.
Response: Thank you for this positive feedback.

Reviewer 2 Report
A great qualitive article on diabetes beliefs and barriers among a much under-studied population. A few, minor comments:
Introduction:
· Any specific statistics available on the % of Fijians or even broadly Australians with diabetes, and/or undiagnosed diabetes? Looking for more context on the severity of diabetes in this population.
Methods:
· Were the interview guides created based on the research questions or were any questions adapted from another source that would need to be referenced?
o Really appreciated seeing the interview guide.
· How many people were recruited, what percentage were interviewed, how many were couples?
Discussion
· Portion control is listed as a cultural norm – this seems like a broad generalization. May consider tempering the language or bring in other factors, such as traditional food types, traditional social norms around eating when invited, etc, that taken all together could be important Fijian dietary points for intervention.
· The term “a ‘communal people” line 222 is unclear – rather, it appears that community is a norm that is prioritized among the Fijian population. Authors make the case in Table 4 of the importance of community for Fijans, especially through church and family, so one option may be to consider using a different term.
Author Response
Reviewer: A great qualitive article on diabetes beliefs and barriers among a much under-studied population. A few, minor comments:
Introduction:
Any specific statistics available on the % of Fijians or even broadly Australians with diabetes, and/or undiagnosed diabetes? Looking for more context on the severity of diabetes in this population.
Response:
Diabetes rates in people in Australia from the Pacific Islands, including Fiji, are higher than the rates in the broader Australian population [Colagiuri et al 2007], however, specific rates for i-Taukei Fijians in Australia are not readily available. Manuscript has been amended as follows to provide context on diabetes prevalence:
People originating from the South Pacific, including Fijians, are disproportionately represented in national diabetes statistics in Australia, where diabetes prevalence was estimated at 7.4% in Australians aged 25 years and over [1]. They are also more likely to be above a healthy weight [2–4] and are at higher risk of diabetes [1,5], with odds of diabetes being 6.3 and 7.2 times higher, after adjusting for age and socioeconomic status, for men and women born in the Pacific Islands compared to the Australian born population [2].
Methods:
Reviewer: Were the interview guides created based on the research questions or were any questions adapted from another source that would need to be referenced?
Really appreciated seeing the interview guide.
Response: The interview guide was adapted from a guide used in a previous study by the authors (FM, KM and DS). The following text has been added at line 66:
The Interview Schedule was adapted from a guide used in a previous study in another population of Pacific Island origin in Sydney by the authors (FM, KM and DS). It has not been previously published.
Reviewer: How many people were recruited, what percentage were interviewed, how many were couples?
Response: Seventeen people initially agreed to participate and 15 were interviewed. Two people who had agreed to participate were shift workers; one in construction and one in after-hours home care. After several attempts to schedule a convenient time for interviews with these 2 potential participants, it was not possible without undue inconvenience for each of them and interviews were not pursued. In total 15 people were interviewed, including 3 married couples who were interviewed as couples. A total of 13 hours of interviews were recorded. The following text has been added:
Fifteen participants were recruited, mean age 49 years (range 26–71 years) (Table 1). Two additional people had agreed to participate but both were shift workers. After several attempts to schedule a convenient time for interviews with these 2 potential participants, it was not possible without undue inconvenience for each of them and interviews were not pursued. In total 15 people were interviewed, including 3 married couples who were interviewed as couples. A total of 13 hours of interview data was recorded and analyzed.
Discussion
Reviewer: Portion control is listed as a cultural norm – this seems like a broad generalization. May consider tempering the language or bring in other factors, such as traditional food types, traditional social norms around eating when invited, etc, that taken all together could be important Fijian dietary points for intervention.
Response: The text has been amended at line 228 – 231, as follows:
For Fijians, feasting is a significant part of many social activities and gatherings where any attempt to practice appropriate portion control may be overwhelmed by other factors, such as bigger being seen as ‘better’ with respect to food portions, family and community influences on eating (where not eating is seen to be not participating) and cultural norms of body image.
Reviewer: The term “a ‘communal people” line 222 is unclear – rather, it appears that community is a norm that is prioritized among the Fijian population. Authors make the case in Table 4 of the importance of community for Fijians, especially through church and family, so one option may be to consider using a different term.
Response: The text has been amended at line 237-239 as follows:
Given these cultural norms and practices as well as ease in being able to adopt an unhealthy lifestyle, a whole community, culturally appropriate approach would appear to be the most practicable method for intervention in the context of the ongoing strong connections within the Fijian diaspora in Australia. Community based approaches can be achieved in several ways.

Reviewer 3 Report
The sample size is relatively small and skewed towards the 45 - 50-year-olds. Adding some statistical data analysis might have added to the scientific soundness of the conclusions.
Author Response
Reviewer: The sample size is relatively small and skewed towards the 45 - 50-year-olds. Adding some statistical data analysis might have added to the scientific soundness of the conclusions.
Response: Participants ranged in age from 26 years to 71 years, with a mean of 50 years for male participants and 46 for female participants, as described in the manuscript. At the 2011 Australian Census, the median age of Fiji born people in Australia was 41 years, with 34.5% in the 45-64 years age group and less than 9% being over 65 years (DIAC 2014).
The age group where prevalence of diabetes begins to increase in the Australian population overall is 45-54 years (AIHW 2016). This fact, together with the age structure of the Fijian community in Australia, support that this sample was not inappropriate for this exploratory study.
This was a qualitative study designed to explore the issues experienced by the i-Taukei Fijian community with respect to diabetes. Therefore, statistical data analysis is neither appropriate nor indicated for this methodological approach. By design, qualitative research samples are small and are not intended to be representative nor to produce data to allow statistical analysis or conclusions. Rather it is intended to facilitate gaining a broad understanding of views and perceptions about a particular topic or issue, in this case, diabetes in the target community, i-Taukei Fijians in Sydney.
The understanding gained from this exploratory study is intended to inform further scientific investigations that would be powered to support statistical analysis and conclusions.
The following text has been added at line 318:
The ages of participants ranged from 26 to 71 years, ensuring representative input across the adult age range, including the 65 years and over group, which accounts for only 9% of the Fiji born population in Australia [7].
References
Department of Immigration and Citizenship (DIAC). Community Information Summary: Fiji-born, 2014. 2014.
Australian Institute of Health and Welfare 2016. Australia’s health 2016. Australia’s health series no 15. Cat no. AUS 199. Canberra: AIHW.
